# The Impact of HPV Diagnosis and the Electrosurgical Excision Procedure (LEEP) on Mental Health and Sexual Functioning: A Systematic Review

**DOI:** 10.3390/cancers15082226

**Published:** 2023-04-10

**Authors:** Michalina Sikorska, Adriana Pawłowska, Anna Antosik-Wójcińska, Aleksandra Zyguła, Barbara Suchońska, Monika Dominiak

**Affiliations:** 1Medical Center of Postgraduate Education, Medical University of Warsaw, Żwirki i Wigury 61, 02-091 Warsaw, Poland; 2Nowowiejski Hospital, Nowowiejska 27, 00-667 Warsaw, Poland; 3Department of Psychiatry, Faculty of Medicine, Collegium Medicum, Cardinal Wyszynski University in Warsaw, Woycickiego 1/3, 01-938 Warsaw, Poland; 4OVIklinika Infertility Center, 01-377 Warsaw, Poland; 51st Department of Obstetrics and Gynaecology, Medical University of Warsaw, 1/3 Starynkiewicza Sq, 02-015 Warsaw, Poland; 6Department of Pharmacology, Institute of Psychiatry and Neurology, Sobieskiego 9, 02-957 Warsaw, Poland

**Keywords:** HPV infection, HPV diagnosis, LEEP procedure, cervical dysplasia, HPV anxiety, HPV depression, psychosocial impact of HPV diagnosis, sexual functioning after LEEP

## Abstract

**Simple Summary:**

This systematic review highlights the impact of HPV diagnosis and the electrosurgical excision procedure (LEEP) on mental health and sexual functioning. Indeed, our results showed that there is a correlation between abnormal cytology results/HPV infection and subsequent treatment interventions (LEEP/LEETZ) and women’s mental health and sexual life. The conclusions of our review underline the importance of sex education and knowledge about sexually transmitted diseases. An improvement in this area could prevent such high transmission of oncogenic HPV. Furthermore, the number of studies evaluating the impact of the procedure (LEEP/LEETZ) and the number of female participants have been very small, thus further research in this direction could be very beneficial. Additionally, psychological support should be provided to any patient who needs it. In this respect, such care should be part of the routine management of patients receiving an HPV diagnosis. Finally, the diagnosis and information about the current state of health should be communicated in an appropriate manner by gynecologists. Proper communication could significantly reduce the patient’s stress and anxiety.

**Abstract:**

The impact of HPV diagnosis and subsequent treatment with the electrosurgical excision procedure (LEEP) on anxiety, depression, psychosocial quality of life, and sexual functioning has not been thoroughly investigated. The aim of this review was to systematically summarize the available knowledge on this topic, according to PRISMA guidelines. Data from observational and intervention studies were analyzed. A total of 60 records were included, of which 50 papers addressed the impact of HPV diagnosis on patients’ psychosocial status, while 10 studies addressed the impact of the implemented LEEP procedure on patients’ mental health and sexual functioning. The results indicated a negative impact of HPV diagnosis on the occurrence of depressive and anxiety symptoms, poorer quality of life, as well as on the sexual functioning of the affected women. The results of the studies to date have not confirmed the negative impact of the LEEP procedure on mental health and sexual life, although more research is needed in this area. It is necessary to implement additional procedures to minimize anxiety and distress in patients receiving a diagnosis of HPV or abnormal cytology and to improve awareness of sexually transmitted pathogens.

## 1. Introduction

Human papillomavirus (HPV) is a micro-sized, envelope-free deoxyribonucleic acid (DNA) virus that infects the cells of the deep layers (basal and basolateral) of the skin or mucosa [1]. HPV infection occurs through contact between infected tissues of the genital area and damaged tissues, including epithelial microdamage, which usually occurs through vaginal, anal, or oral sex, as well as intercourse with accessories [2]. Most HPV infections are self-limiting, asymptomatic, and often remain undiagnosed, especially in young adults.

Oncogenic types of HPV called high-risk HPV (HR HPV), mainly HPV 16 and 18, tend to cause most precancerous conditions and cancers of the cervix, as well as of the vulva, vagina, anus, penis, oral cavity, and throat. Persistent HR HPV infection is a major risk factor for the development of HPV-associated cancers of the anogenital region, as well as the head and neck area. Infection with non-oncogenic types of HPV called low-risk HPV (LR HPV), mainly HPV 6 and 11, can cause genital warts, including condylomata, and recurrent respiratory papillomatosis. However, they are not risk factors for cancers of viral etiology [3].

Methods that reduce the risk of developing cervical cancer include primary prevention, such as HPV vaccination, and health education through the use of condoms, reducing the number of sexual partners, and general health behaviors, such as smoking cessation; as well as secondary prevention with screening tests [4].

One can venture to say that in Poland sex education is poor and does not meet contemporary expectations or the needs of adolescents and young adults. The elderly were in a comparable situation at the time of sexual initiation and later. As a result, there is no widespread public awareness of the health and emotional consequences of sexual contact without physical protection (condom) and, when possible, immunological protection (HPV vaccination). The only information about the consequences of sexual imprudence concerns the potential risk of unplanned pregnancy. What is missing is information on the risk of sexually transmitted infections, such as HIV, syphilis, hepatitis B, hepatitis C, chlamydiosis and parasitic diseases (cysticercosis, toxoplasmosis), and HPV infection [5]. Nevertheless, as these infections can lead to infertility or the development of cancers in the genital area, this seems to be a very important health topic. Often, it is only in the case of abnormal results during check-ups that women try to make up for their knowledge in this area, in a short period of time. The above situation can exacerbate stress, affecting relationships with a partner and their sex life, sometimes even eliminating it completely. The main cause of this increased stress is a lack of knowledge about safe sexual behavior and the need to perform a wide range of tests for sexually transmitted infectious diseases after each change of partner.

Persistent HPV HR infection in the cervix can lead to the formation of high-grade squamous intraepithelial lesions (HSIL) or cervical intraepithelial neoplasia (HSIL CIN-2/CIN-3) lesions that precede the onset of cervical cancer. Women with histologically confirmed high-grade lesions (CIN2+) in the cervical mucosa are typically recommended to undergo surgical treatment, which involves the removal of the affected area using a procedure known as the loop electrosurgical excision procedure (LEEP) or large loop excision of transformation zone (LLETZ). The stress of being diagnosed with a disease closely related to a woman’s sexual intercourse and her partners, the prospect of surgery, the altered anatomical relations within the cervix after surgery, and the change in the location of nerve endings can all take a toll on patients’ overall well-being, mental state, and perception of sexual satisfaction.

Studies have indicated that emotional reactions to a HPV infection test among women in the general population are also influenced by socioeconomic factors. Women with a lower education level, unemployed, younger age, unmarried, or not cohabiting may be more likely to experience adverse psychological reactions (feelings of shame, anxiety, stigma, and worry) associated with a positive HPV infection test. Information regarding a positive HPV test result in Poland is usually preceded by an abnormal gynecological cytology result. This is often accompanied by anxiety, fear of cancer, feelings of harm and injustice, and a search for the culprit of HPV infection. Currently, various reports related to the impact of a positive HPV test result on patients’ emotional reactions are available. Studies indicated that women receiving such a diagnosis are at risk of anxiety and depression [6,7,8,9,10]. However, it seems that this negative impact may be transient and diminishes in the following months [6,7,8,9,10]. On the other hand, one can also find studies that do not support the transient nature of this problem, indicating anxiety and mood problems long after diagnosis [9,11].

At present, HPV-related genital precancerous lesions appear frequently. While LSIL (low-grade) lesions are likely to resolve within a few years, HSIL (high-grade) lesions can cause a well-founded fear of progression and tend to recur, especially during the first few years after excitotoxic treatment. Consequently, they may also raise concerns about the consequences on sexual health. Referral for in-depth diagnostics, including colposcopic evaluation with histopathological verification is an integral part of providing medical services for the diagnosis and treatment of precancerous conditions of the cervix. The psychosocial impact of a positive HPV test can result, not only in psychological stress, but also significantly affect treatment outcomes through compliance. The World Health Organization (WHO) recommends counseling as an interpersonal communication strategy between a health professional and the female patient, to learn more about HPV and cervical cancer prevention and to give room for conversations about intimate topics, such as sexuality, disease, and death, and thus to encourage women to take preventive measures [12]. Thus, many aspects of HPV infection can interfere with women’s mental and sexual health, from the moment of receiving a diagnosis of infection with a particular strain of HPV, through the treatment process, to follow-up visits. Regarding the treatment procedure (LEEP/LLETZ), there is a lack of consensus on the impact on patients’ sexual life and well-being. The study by Inna et al. [13] proved that excisional treatment of cervical lesions was associated with a decrease in overall sexual satisfaction. On the contrary, Gaurav et al. noticed a significant improvement in symptoms of dyspareunia and postcoital bleeding after the LEEP procedure [14]. In turn, three other researchers found no differences in sexual functioning and mental well-being between women treated and not treated with LEEP [15,16,17].

Given the relevance of the topic of HPV infection and its health consequences, we aimed to systematically review the literature and summarize the conclusions from the available studies.

## 2. Aim

The aim of this study was to review the available literature in terms of (1) the impact of abnormal cytology results/HPV infection on women’s mental health and sexual life; and the impact of (2) subsequent treatment interventions (LEEP/LEETZ) on women’s mental health and sexual life.

## 3. Methods

On 26 September 2022, a search for articles was conducted in the MEDLINE, Embase, PsycINFO, and BIOSIS databases, with the aim of reviewing the impact of abnormal cytology results/HPV infection on women’s mental health and sexual life, and the impact of subsequent treatment interventions (LEEP/LEETZ) on women’s mental health and sexual life. Terms targeting anxiety, mood disorders, and sexual dysfunction related to a HPV-positive test or LEEP procedure were used, employing medical subject headings (MeSH), text words, and keywords. The following combinations of words were used: (‘mood disorders’ OR ‘major depression’ OR ‘depression’ OR ‘anxiety’ OR ‘HPV anxiety’ OR ‘HPV depression’ OR ‘HPV diagnosis’ OR ‘sexual anxiety’ OR ‘sexual dysfunction’ OR ‘HPV sexual dysfunction’ OR ‘cervical dysplasia’ OR ‘HPV positive test’ OR ‘psychosocial impact of HPV’) AND (‘sexual life after LEEP’ OR ‘LEEP anxiety’ OR ‘LEEP sexual dysfunction’ OR ‘LEEP procedure’ OR ‘LEEP’ OR ‘sexual function after LEEP’ OR ‘painful intercourse after LEEP’). The findings were organized in accordance with the recommended format of the preferred reporting items for systematic reviews and meta-analyses (PRISMA) guidelines (see Figure 1). In accordance with the PRISMA guidelines, the review protocol was recorded in the PROSPERO database (Prospero-ID: CRD42022383640, December 2022). A comprehensive search of the literature was conducted, and three researchers (MS, MD, and AZ) worked independently to locate and identify relevant sources. The reviewers (MS, MD, and AZ) then independently screened the titles and abstracts of the articles, to determine their potential relevance. The full text of all eligible or provisional studies was obtained, to determine whether a study met the eligibility criteria. All disagreements between reviewers (MS, MD) were resolved through discussions, to reach a consensus. The articles were included for analysis if the subject of the search appeared in the article title or abstract. At the outset, research conducted on animal subjects, book chapters, systematic reviews, brief assessments, encyclopedic entries, summaries of conference presentations, written exchanges, short written communications, dialogues, editorials, letters, brief records, and concise surveys were excluded. The investigation was limited to publications written in the English language. Furthermore, a process was conducted to remove duplicate results. Following this, the abstracts of the retrieved articles were analyzed, and studies that examined how an abnormal cytology result, HPV infection, and subsequent medical interventions impact the psychological well-being and sexual life of women were isolated. Other publications that did not meet the above conditions or were not related to PICO questions (Appendix A) were excluded. A summary of the article search results is provided in Figure 1. In the analysis, information presented in the scientific works, including the authors; publication year; country of origin; study design; sample sizes for both the study and control groups; characteristics of the study and control participants, such as gender, mean age, diagnosis, and treatment; type of tests administered; and the results obtained from professional scales utilized to assess the incidence of depression were taken into account. Further details regarding the results are provided in the Results section.

Out of the 1351 records initially retrieved, 374 were found to be ineligible based on the inclusion criteria. After removing 374 duplicate articles, 609 articles were deemed eligible. Upon reviewing the titles and abstracts, an additional 1291 publications were excluded for the following reasons: involving animal models, being a review paper, or having a prospective observational design format. Other exclusion criteria included studies that did not describe patients with HPV diagnosis or an abnormal cytology result, related medical interventions, and psychological impact on mental health and sexual life. However, 60 articles met the inclusion criteria, of which two concerned the same clinical trial. The results are shown in Figure 1.

## 4. Results

### 4.1. Study Characteristics

We reviewed 609 publications on the impact of an abnormal cytology result, HPV infection, and subsequent medical interventions on women’s mental health and sexual life. Sixty records met the inclusion criteria (Appendix A). Fifty-five of these 60 papers were observational studies, including 26 prospective observational studies, 20 cross-sectional observational studies, three cross-sectional qualitative studies, two retrospective observational studies, one comparative observational study, one comparative qualitative study, one observational cohort study, and one prospective/retrospective observational study. The average duration of the above observational studies was 3 months (range 1–112 weeks). The studies included participants who had the following characteristics: (1) women with a positive HPV test (n = 22); (2) women suffering from genital lesions linked to HPV infection (n = 11); (3) women with abnormal cervical cytology (n = 10); (4) women with low-grade cytological abnormalities (n = 7); (5) women undergoing LEEP (n = 9); (6) women diagnosed with HR HPV infection (n = 3); (7) women diagnosed with a borderline Pap smear (n = 3); (8) women undergoing a Pap smear, viral typing, or colposcopy (n = 2); (9) women diagnosed with CIN-2 or CIN-3 lesions and scheduled for LEEP (n = 1); and (10) women who reported sexual dysfunction following LEEP procedure (n = 6). Our systematic review also included four interventional studies, including three randomized controlled studies, and one single-arm non-randomized interventional study. The total number of patients included in the research was 32,476, with a group size ranging from 12 to 3753 participants.

The participants involved in the research had an abnormal cytology result or had received an HPV-positive test result. In order to understand the impact of an abnormal cytology result, HPV infection, and subsequent medical interventions on women’s mental health and sexual life, participants were screened and examined according to Diagnostic and Statistical Manual of Mental Disorders (DSM) criteria, using the Hospital Anxiety and Depression Scale (HADS) [15,19,20,21,22,23,24,25,26,27,28], Beck Depression Inventory (BDI) [29,30], Beck Anxiety Inventory (BAI) [29,30,31], Symptom Checklist-Revised (SCL-90) [32], Psychosocial Effects of Abnormal Pap Smears Questionnaire short-form (PEAPS-Q) [11,24,33,34,35], State-Trait Anxiety Inventory (STAI) [6,7,8,9,10,11,16,33,35,36,37,38,39,40], Female Sexual Function Index (FSFI) [15,17,31,39,41,42], HPV Impact Profile (HIP) [7,9,21,25,26,43,44], Patient Health Questionnaire-4 (PHQ-4) [45], General Health Questionnaire (GHQ-12) [6,10,11,33,46], Cervical Screening Questionnaire (CSQ) [11,22,33,35,36,38], Specific questionnaire for Condylomata Acuminata (CECA) [44]; Courtauld Emotional Control Scale (CECS) [21,23], Fear of Progression Questionnaire (FoP-Q) [47,48], Arizona Sexual Experiences (ASEX) [29,41], The 12-item Short Form Survey (SF-12) [7], Short-Form-36 (SF-36) [32,35], Visual Analog Scale (VAS) [7], EuroQol-5 Dimension (EQ-5D) [7,44,49], Symptom Checklist of Sexual Function (SCSF) [50], Process Outcome Specific Measure (POSM) [27], Intensive Care Psychological Assessment Tool (IPAT) [51], International Index of Erectile Function (IIEF) [46], Cervical Dysplasia Distress Questionnaire (CDDQ) [47,48,51], Ask Suicide-Screening Questions (ASQ) [51], Illness Attitude Scales (IAS) [46], Brief Illness Perception Questionnaire (BIPQ) [36,37], Health-Related Quality of Life (HRQoL) [8,19], Index of Sexual Satisfaction (ISS) [21,23], Illness Perception Questionnaire (IPQ-R) [36,37], Breast Cancer Worry Scale [9], Psychosocial Adjustment to Illness Scale-SR (PAIS-SR) [34], Spiritual and Religious Attitudes in Dealing With Illness (SpREUK) [21,23], Female Sexual Distress Scale-Revised (FSDS-r) [15], Cognitive Behavioural Assessment (CBA-20) [52], Positive and Negative Affect Schedule (PANAS) [40], Experiences in Close Relationship Scale-Short Form (ECR-S) [23], Brief Index of Sexual Function for Women (BISF-W) [46,52], Satisfaction Profile (SAT-P) [52], Functional Assessment of Chronic Illness Therapy- Cervical Dysplasia (FACIT-CD) [19,20], Sexual Activity Questionnaire (SAQ) [47,48], Intolerance of Uncertainty Scale (IUS) [40], Need for Closure Scale (NFCS) [40], Mishel Uncertainty in Illness Scale (MUIS) [40], HPV Knowledge Questionnaire (HPVQ) [23], Papanicolaou Exam Knowledge Questionnaire (PEKQ) [23], and Revised Dyadic Adjustment Scale (RDAS) [23]. Depression, anxiety, general quality of life, and sexual function, as well as the effect of therapy on quality of life, mental state, and sexual function were assessed through the mentioned scales. In addition, 18 studies used non-validated questionnaires, surveys, or qualitative face-to-face interviews to assess mental status and quality of sexual life [13,14,50,51,53,54,55,56,57,58,59,60,61,62,63,64,65,66].

This review includes 59 studies, where authors investigated the effect of HPV diagnosis on anxiety, depression, and psychosocial quality of life (Appendix A), while in 10 publications, patients were evaluated by measuring the impact of the loop electrosurgical excision procedure (LEEP) on anxiety/depression, quality of life, and sexual function (Appendix A). Among the studies, different types of medical intervention were used. A screening following cervical cytology (including abnormal results) was used in 31 studies. The HPV test constituted a diagnostic tool in 26 studies. Female patients with genital warts linked to HPV were included in nine publications. Patients who had undergone the loop electrosurgical excision procedure were described in 10 research papers, and women who were referred to diagnostic colposcopy were involved in three studies. Most of the studies were performed in the United Kingdom (13), Italy (6), Australia (4), Canada (4), Austria (3), Sweden (3), and Turkey (3), followed by China (2), Israel (2), Lebanon (1), Portugal (2), with single representations from Brazil (1), Colombia (1), Denmark (1), Finland (1), France (1), Germany (1), Greece (1), India (1), Iran (1), Ireland (1), New Zealand (1), Norway (1), Serbia (1), South Korea (1), Taiwan (1), Thailand (1), and the USA (1).

### 4.2. Quality Assessment

The observational studies were classified using the ROBINS-E tool. Fourteen of the fifty-six included studies were rated as ‘low risk of bias’ [10,13,15,22,24,25,26,29,30,34,36,40,47,48], sixteen as ‘some concerns’ [8,9,11,14,17,19,32,39,44,49,52,53,54,55,56,67], twenty as ‘high risk’ [6,7,16,27,31,33,35,38,41,42,43,45,46,50,51,57,58,59,60,61], and five as ‘very high risk’ [62,63,64,65,66]. The risk of bias in interventional studies was evaluated using two tools: the RoB 2 tool, which is a revised tool for assessing the risk of bias in randomized trials; and the ROBINS-I tool, which is used for assessing the risk of bias in non-randomized studies and interventions. Among the four interventional studies examined, one was determined to have a ‘low risk of bias’ [40], while the other two were categorized as having ‘some concerns’ [22,28]. Using the ROBINS-I tool, the risk of bias in one non-randomized controlled study was assessed and found to be ‘moderate risk’ [20]. The risk of bias for all studies is presented in Appendix A.

### 4.3. Impact of HPV Diagnosis on Anxiety, Depression, Quality of Life, and Sexual Functions

Of the 49 studies assessing mental and sexual health among women with HPV infection, 45 were observational studies, of which 30 used validated tools to assess anxiety, depressive symptoms, quality of life and sexual function, and 15 were qualitative studies using unvalidated questionnaires, surveys, or interviews (Appendix A).

Forty-six studies assessed the impact of the HPV diagnosis on depressive or anxiety symptoms: nine on quality of life, and 15 on sexual function (Appendix A).

Nineteen studies simultaneously assessed more than one of the functional aspects mentioned above, and two concerned well-being in relation to the way medical personnel conveyed information. The following groups were considered: women positive for HPV (n = 22); women with genital lesions linked to HPV (n = 11); women with abnormal cervical cytology (n = 10); women with low-grade cytological abnormalities (n = 7), women diagnosed with high-risk HPV (n = 3); women diagnosed with a borderline smear (n = 3); and women undergoing a Pap smear, viral typing, or colposcopy (n = 2). Of the 44 studies assessing the impact of HPV infection on anxiety and depressive symptoms, 41 (93%) confirmed a negative impact of the diagnosis on these symptoms (29/32; 90% of quantitative and 11/12; 92% of qualitative studies). Patients also reported feelings of fear and worry [53,58,62,63], as well as shame and guilt [54,62,65]. HPV infection also had a negative impact on quality of life according to 67% of studies (5/8; 62% of quantitative and 1/1; 100% of qualitative studies). Additionally, among the 15 studies focused on the relationship between positive HPV testing and its impact on sexual dysfunctions, the majority of studies (87%) confirmed this unfavorable relationship (9/10; 90% of quantitative and 4/5; 80% of qualitative studies). Patients reported sexual concerns in particular, such as sexual desire, arousal, genital response, orgasmic experience, and satisfaction from orgasm [29,41,46].

### 4.4. Impact of the LEEP Procedure on Anxiety, Depression, Quality of Life, and Sexual Functions

The impact of the loop electrosurgical procedure on anxiety, depression, quality of life, and sexual functions was assessed in 10 studies (Appendix A). Seven of these were quantitative studies using validated instruments to assess mental state and sexual function, while three were qualitative studies. Most of these (8/10) studies were of good quality (‘low risk of bias’/‘some concerns’) [13,14,15,17,24,30,39,55], and two were assessed as ‘high risk of bias’ [16,42]. Four studies addressed the issue of mental health after the LEEP procedure [15,24,30,62], of which only one study found a negative effect of LEEP on anxiety symptoms [30] (1/4, 25%), while 75% (3/4) did not confirm these findings [15,24,39]. In total, six out of 10 studies, or 60% (5/7, 71% of quantitative studies and 1/3, 33% of qualitative studies), did not confirm a negative impact of the LEEP procedure on sexual functions [14,15,16,17,24,39], while four studies, or 40% (2/7, 28% of quantitative studies and 2/3, 67% of qualitative studies), found such an association [13,30,42,55]. Some participants reported that symptoms of dyspareunia and postcoital bleeding, which indirectly affected sexual health, showed significant improvement post-LEEP procedure [14]. In another study, participants reported symptoms of altered or loss of cervical sensation during sex (numbness); pain with vaginal penetration; decreased lubrication; depression; loss of interest [30], arousal, desire, and self-confidence; as well as issues with partners after the LEEP procedure [13,42]. On the other hand, Hellsten et al. [16] stated that women who underwent the LEEP procedure, compared to the women who had not experienced this procedure, did not differ in their sexual functioning. Very similar results were obtained in the study conducted by Michaan et al. and Serati et al. [15,17].

## 5. Discussion

This review summarizes the existing literature on the impact of an abnormal cytology result, HPV infection, and subsequent medical interventions on women’s mental health and sexual life. To the best of our knowledge, this is the first systematic review covering this topic. The available literature suggests that there was a broad diversity in the methodology used in the conducted studies. Most of the studies performed to date focused on the overall psychosocial impact, including anxiety, anger, disappointment, depression, fear, feelings of shame and guilt, and sexual functioning.

The fear and anxiety caused by abnormal Pap smear results or positive HPV tests are largely due to the screening program itself [57]. Without screening, women would not know about asymptomatic precursors (CIN2+) to cervical cancer. Without screening, fewer women would be exposed to abnormal findings, but more women would develop cervical cancer [68]. The sensitivity of HPV testing surpasses that of cervical cytology (Pap test), but the specificity is lower. Primary HPV testing can reduce the incidence of cervical cancer more effectively than cervical cytology screening, albeit with a higher number of positive screening results. Given the negative emotional impact of screening, more specific tests are needed. The positivity rate of the HPV test, the referral rate, and detection rates of CIN3+ can be influenced by the number of HPV types included in the test [69]. A previous study revealed that only six HPV types (16, 18, 31, 33, 45, or 52) accounted for 85% of invasive cervical cancer cases, whereas the other eight HPV types detected in a 14-type HPV DNA test were found in only 1.5% of invasive cervical cancer cases [70]. It is worth noting that not all cases of CIN3 will progress to cancer, and only a minority of CIN2 cases do. Research indicates that if left untreated, only 30% of large CIN3+ lesions will develop into cervical cancer over 30 years [71], indicating that a considerable number of women with these conditions will undergo biopsy and, potentially, LEEP treatment needlessly. The primary objective of cervical cancer screening is not to detect as many CIN3+ cases as possible but to prevent as many cases of cervical cancer as possible, while balancing the benefits and harms [72].

Despite such varied methodologies, it can be summarized that receiving information about abnormal cytology results or an active HPV infection has a negative impact on women’s mental health and sexual life. Among studies focused on measuring the impact of HPV diagnosis on anxiety or depression, as many as 89% confirmed such a negative impact (Appendix A). The impact of HPV diagnosis on quality of life was not as pronounced; it was confirmed in 67% of studies. Nevertheless, the analyzed studies demonstrated a large impact on the lifestyle, habits, and social life of women with cervical abnormalities [6,7,8,9,10,19,21,27,28,33,34,36,37,38,39,40,43,44,47,49,50,52,53,56,57,58,59,60,62,63,64,65]. As regards the impact on sexual function and quality of sexual life, 87% of the studies analyzed confirmed this negative effect. An HPV-positive result can lead to the loss of sexual desire, or to the transformation of sexuality into an unpleasant experience [34,38,48,49,51,60]. Ferenidou et al. described difficulties in sexual life that can occur after a diagnosis of HPV, such as impaired sexual desire and pain during intercourse [50]. Feelings of concern and worries were mostly strongly felt in the study population of women with genital warts, who were more concerned with the risk of transmission and the impact on their partners [25,48,65]. An important issue raised in the study of Santos et al. [23,73] was the correlation of age and satisfaction with sexual functionality. It was noted that older women reported greater sexual dissatisfaction six months after HPV diagnosis [23]. A study by Alay et al. also assessed whether specific HPV genotypes affect anxiety and sexual satisfaction in women. It was found that the detection of HPV 16 or 18 infections, the etiologic agents of 70% of cervical cancers, had a particularly adverse effect on women’s total sexual functioning score (FSFI) and desire score compared to other genotypes [31].

It is important to emphasize that the women participating in the studies described shame, especially when they were informed in advance that HPV infection occurs mainly through unprotected sexual intercourse. Since sexual intercourse is still considered a taboo topic for most of society, in particular accidental intercourse, the women felt humiliation. McCaffery et al. found negative social and psychological implications associated with a positive HPV test result. Women were found to be stressed about disclosing their results to others [62]. In addition, some of the patients described difficulties in establishing new sexual relationships as a result of knowing they were infected [54,61,62]. Patients who also suffered from HPV-related genital warts described a significantly greater sense of shame than HPV-positive individuals without warts [25,26,44,48,49]. Additionally, patients often expressed fear of becoming pregnant upon receiving a positive HPV test result, although there is no causal relationship between the two. Nonetheless, this topic is of significant importance and deserves a dedicated paper to address it thoroughly.

As regards the impact of the LEEP procedure on anxiety or depression and quality of life, we identified only four studies addressing this issue. These studies did not indicate any particular negative impact of the LEEP procedure on mental health. However, more research is needed, as studies addressing this issue are lacking. In contrast, more work has been done on the impact of the LEEP procedure on the quality of sexual life. We found fairly divergent results, with a predominance of no adverse effects of the LEEP procedure on sexual function, as indicated by 60% of the analyzed papers. The study conducted by Gaurav et al. indicated an overall improvement in sexual satisfaction, by minimizing the occurrence of persistent dyspareunia symptoms and postcoital bleeding in female patients [14]. The beneficial effect of this procedure was also noted by Plotti et al. [39]. Patients who underwent electroconization showed a better overall anxiety status, as well as statistically significant differences in sexual satisfaction [39]. On the other hand, three research groups [15,16,17] observed no differences in sexual functioning between women who underwent the LEEP procedure and those who did not. There were also reports of adverse effects, such as loss of cervical sensation during intercourse (numbness); pain during vaginal penetration; decreased lubrication; loss of interest, arousal, desire, and confidence; as well as trouble with partners [42]. Another study found that women experienced small, but statistically significant, decreases in overall sexual satisfaction, vaginal elasticity, and satisfaction with orgasm [13]. Sparić et al. also noted that 27.4% of women were less interested in sexual intercourse after the LEEP procedure compared to before the procedure [30]. Moreover, sexual functioning may also have been influenced by the age of the women, as described by Giannella et al. [55]. The study showed that postmenopausal women experienced significant psychological changes concerning the impact of cervix disease, their body image, interpersonal relationship with their partner, and sexual health quality after undergoing the LEEP procedure. Such a relationship could indicate a greater negative impact of the LEEP procedure in postmenopausal women [55]. However, it is difficult to determine whether this adverse effect of the LEEP procedure on patients’ sexual satisfaction is clinically significant. For example, in the study conducted by Giovannetti, patients also reported a lack of satisfaction with preoperative and postoperative care. Furthermore, the complaints also concerned the level of information about the condition and the LEEP procedure that was performed. In addition, women did not feel comfortable when discussing sexual problems with healthcare professionals [42].

An additional important issue that is worth addressing is the education and socio-economic factors of those receiving a positive HPV test result. Women with lower education and lower socio-economic levels showed significantly higher levels of stress and anxiety. This fact may be related to the better education, understanding, and knowledge of the causes and consequences of the virus infection in women classified with a higher socio-economic level [7,41,59]. Such a hypothesis may be confirmed by the study by Mortensen et al., which showed that women expressed a fear of having cancer that was not proportional to the degree of cervical dysplasia, but to their level of knowledge on the topic [63]. Moreover, Monsonego et al. conducted a study that showed the importance of proper psychological care after receiving abnormal results. The women mostly complained about the level of psychological support after receiving an abnormal result and also indicated that they did not receive a sufficient amount of information from the doctor who communicated the result to them [58]. Similary, Dellino et al. conducted a monocentric survey on HPV risk information and found that almost half (48%) of the participants expressed the need for qualified personnel to provide comprehensive and standardized communication. The study also revealed that the use of the terms ‘high risk’ and ‘low risk’ in the current reporting system increased feelings of anxiety and distress in 63% of patients. Additionally, over a third (34.5%) of women surveyed believed that psychological support should be incorporated into the screening process, to assist women with HPV lesions [74]. Interestingly, there were also studies that proved that anxiety is not dependent on the type of HPV diagnosis [45]. Such a result may also be related to a lack of knowledge about the accuracy of the diagnostic tests received and their sensitivity and specificity. Marlow et al. and McCaffery et al. highlighted in their work the importance of proper communication between the patient and healthcare professionals [6,56]. Anxiety and distress have been described in women who were HPV positive, even 12 months after diagnosis. The authors noted that it is extremely important to implement further diagnosis and retesting, as in some women the virus may have been eliminated from the body. Without such knowledge, they may continue to suffer the negative effects of stress, affecting their mental health [6]. Importantly, Gaurav et al. and Plotti et al. described a significant decrease in anxiety status in HPV-cleared patients after cervical treatment [14,39]. The importance of proper communication was also demonstrated in studies in which researchers noted a correlation between the consulting style and the psychological response to the HPV diagnosis [56].

Overall, the analysis of the studies indicated a huge deficit in women’s knowledge about sexually transmitted diseases, especially about HPV and its oncogenicity. This highlights the need for an increase in sex education in society, especially concerning knowledge about these diseases and routes of transmission. In addition, information on HPV vaccination could reduce the risk of infection and thus the risk of cervical cancer. It can be assumed that such knowledge would result in a reduced incidence of unprotected intercourse or accidental intercourse. Moreover, full information about the available and emerging new treatments, both surgical and non-surgical (such as vaccination, probiotic treatment) [75], would allow patients to reduce their anxiety and stress associated with diagnosis. Women who were diagnosed with HPV or cervical dysplasia or who underwent LEEP surgery also complained of a lack of adequate psychological support and inadequate provision of information about their situation from healthcare professionals. Therefore, staff training is also an area for further improvement. Although we did not come across any specific guidelines in our research, we believe that offering joint education for both the patient and her partner could be beneficial. This approach would ensure that both parties in the relationship receive sufficient information and are well-informed. We believe that providing psychological care after receiving a positive HR HPV test and throughout the LEEP procedure could have significant benefits for the patient’s well-being. Coping with a positive HPV diagnosis can be challenging, and psychological care can help women manage the stress and anxiety associated with diagnosis.

Training midwives in this area could be a prudent step. Moreover, providing informational and educational brochures to patients, allowing them to learn about the topic in the comfort of their homes, could also be a valuable solution. Furthermore, sexological care can be important for women, who may experience changes in their sexual life or relationships after diagnosis and treatment. This type of care can provide women with information and resources to address sexual concerns and improve their sexual function and satisfaction.

### Study Limitations

Due to the differences in the outcome of mental health and sexual functioning measures, it was not possible to conduct a meta-analysis. Given that there are disparities between countries associated with health and welfare systems, we cannot conclude with certainty that the insufficient number of studies from other parts of the world did not affect the results of this study. In addition, we only included studies written in English.

## 6. Conclusions

The results of this systematic review show that receiving a positive HPV test or abnormal cytology result has a negative impact on women’s mental health and sexual function. Moreover, although studies focused on sexual well-being and functioning after the LEEP procedure showed divergent results, the majority of studies did not confirm negative consequences of the procedure. Nevertheless, more randomized trials are needed to investigate this topic further.

In addition, the studies included in this review provide insights into patients’ perceptions of their situation. There is still much work to be done in sexual education, including providing information about sexually transmitted diseases. It seems crucial to provide patients with professional information about their condition or planned medical procedures, which could significantly reduce their stress and anxiety.

It is crucial to recognize that each woman’s experience with HPV and the LEEP procedure may differ, and their requirements for psychological and sexological care may vary. Thus, healthcare providers should offer personalized care that meets each woman’s unique needs and preferences, to ensure optimal outcomes.

This review highlighted the need to implement additional procedures to minimize anxiety and distress in patients receiving a diagnosis of HPV or abnormal cytology, and to improve awareness of sexually transmitted pathogens. The impact of the LEEP procedure on mental health and sexual life has not been confirmed by the current studies, and further research in this area, on a larger group of women undergoing this procedure, is necessary. In addition, it would be worth expanding the study to a larger group of people, after implementing the aforementioned broad psychological and sexological support, along with extending information and educational activities. Studies focusing on concerns about getting pregnant in women with a positive HR HPV test could also be valuable.

## Figures and Tables

**Figure 1 cancers-15-02226-f001:**
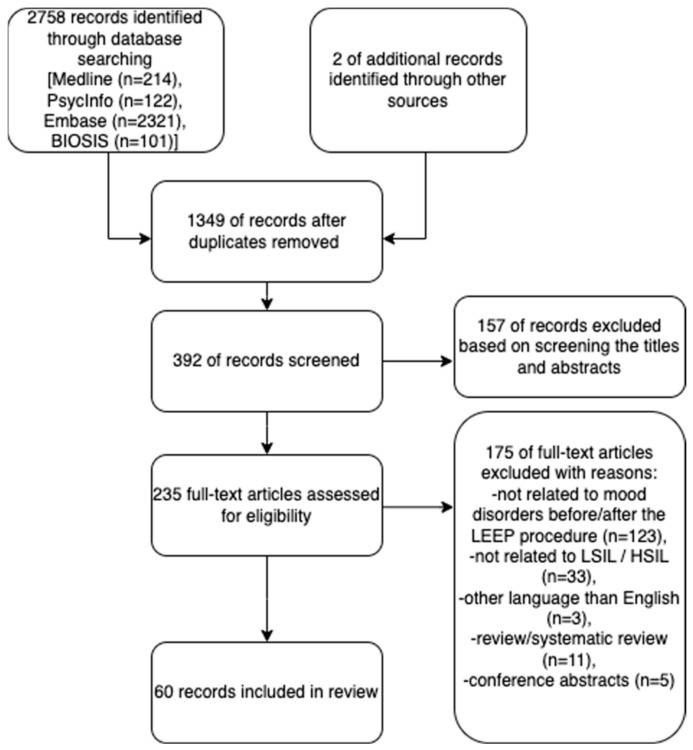
Prisma flow diagram [18] for the studies on the impact of an abnormal cytology result, HPV infection, and subsequent medical interventions on women’s mental health and sexual life.

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
