# Peer review of "The Impact of HPV Diagnosis and the Electrosurgical Excision Procedure (LEEP) on Mental Health and Sexual Functioning: A Systematic Review"

_cancers, 2023, doi:10.3390/cancers15082226_

Round 1

Reviewer 1 Report

Dear Authors,

Thank you for allowing me to review your manuscript.

The study is very interesting and I really congratulate you on this research.

The introduction is well documented and the purpose of the study is clearly expressed.

In the results section some correlations are always a good point.

The discussion and conclusions are well written.

But I have doubts, that is, did the patients willingly agree to undergo such long questionnaires?

Could you provide us with a copy of the informed consent you have given to these patients?

Have you made sure that these psychological measurement tools are standardized and the most innovative?

Did this not interfere with clinical practice?

What do you think needs to change so that a diagnosis of HPV does not affect the marital relationship and the mental health of the patient?

You have described the need for psychological support, but at what time and in what cases should it be involved in the management of the patient? 

In clinical practice, patients report being afraid of becoming pregnant after the history of HPv, but things are absolutely unrelated. Have you also investigated the relapse of these sexual dysfunctions also on the reproductive outcome? (That is, for example, that women who would have wanted a child avoided looking for it after being diagnosed with HPV).

Finally, for a multidisciplinary approach we propose to mention:

-DOI: 10.1186/s13027-022-00465-9

-DOI: 10.3390/jpm12091387 

Author Response

Thank you for your comment. Please find the answers to the questions below.

But I have doubts, that is, did the patients willingly agree to undergo such long questionnaires?

Could you provide us with a copy of the informed consent you have given to these patients?

Response: Our manuscript is a systematic review, which means we did not have any contact with patients during the study. However, the studies we reviewed, at least all of those rated as high quality, reported that patients gave written informed consent.

Have you made sure that these psychological measurement tools are standardized and the most innovative?

Response: While evaluating the research's quality, we considered what tools were used in the study. Most of the studies used standardized tools (participants were screened and examined according to the Diagnostic and Statistical Manual of Mental Disorders (DSM) criteria, using the Hospital Anxiety and Depression Scale (HADS) [15,19,28,20–27], Beck Depression Inventory (BDI) [29,30], Beck Anxiety Inventory (BAI) [29–31], Symptom Checklist-Revised (SCL-90) [32], Psychosocial Effects of Abnormal Pap Smears Questionnaire short-form (PEAPS-Q) [11,24,33–35], State‐Trait Anxiety Inventory (STAI) [6,7,37–40,8–11,16,33,35,36], Female Sexual Function Index (FSFI) [15,17,31,39,41,42], HPV Impact Profile (HIP) [7,9,21,25,26,43,44], Patient Health Questionnaire-4 (PHQ-4) [45], General Health Questionnaire (GHQ‐12) [6,10,11,33,46], Cervical Screening Questionnaire (CSQ) [11,22,33,35,36,38], Specific questionnaire for Condylomata Acuminata (CECA) [44]; Courtauld Emotional Control Scale (CECS) [21,23], Fear of Progression Questionnaire (FoP-Q) [47,48], Arizona Sexual Experiences (ASEX) [29,41], The 12-item Short Form Survey (SF-12) [7], Short-Form-36 (SF-36) [32,35], Visual Analog Scale (VAS) [7], EuroQol-5 Dimension (EQ-5D) [7,44,49], Symptom Checklist of Sexual Function (SCSF) [50], Process Outcome Specific Measure (POSM) [27], Intensive Care Psychological Assessment Tool (IPAT) [51], International Index of Erectile Function (IIEF) [46], Cervical Dysplasia Distress Questionnaire (CDDQ) [47,48,51], Ask Sui-cide-Screening Questions (ASQ) [51], Illness Attitude Scales (IAS) [46], Brief Illness Per-ception Questionnaire (BIPQ) [36,37], Health-Related Quality of Life (HRQoL) [8,19], Index of Sexual Satisfaction (ISS) [21,23], Illness Perception Questionnaire (IPQ-R) [36,37], Breast Cancer Worry Scale [9], Psychosocial Adjustment to Illness Scale-SR (PAIS-SR) [34], Spiritual and Religious Attitudes in Dealing With Illness (SpREUK) [21,23], Female Sexual Distress Scale-Revised (FSDS-r) [15], Cognitive Behavioural Assessment (CBA-20) [52], Positive and Negative Affect Schedule (PANAS) [40], Experiences in Close Rela-tionship Scale-Short Form (ECR-S) [23], Brief Index of Sexual Function for Women (BISF-W) [46,52], Satisfaction Profile (SAT-P) [52], Functional Assessment of Chronic Illness Ther-apy- Cervical Dysplasia (FACIT-CD) [19,20], Sexual Activity Questionnaire (SAQ) [47,48], Intolerance of Uncertainty Scale (IUS) [40], Need for Closure Scale (NFCS) [40], Mishel Uncertainty in Illness Scale (MUIS) [40], HPV Knowledge Questionnaire (HPVQ) [23], Papanicolaou Exam Knowledge Questionnaire (PEKQ) [23], Revised Dyadic Adjustment Scale (RDAS) [23]). In addition, 18 studies used not valid questionnaires, surveys or qualitative face-to-face interviews to assess mental status and quality of sexual life [13,14,50,51,53–66] (230-263).

Did this not interfere with clinical practice?

Response: As far as clinical practice is concerned, we have no evidence to suggest that participation in a clinical trial, such as completing a questionnaire, would interfere with the clinical process. High-quality studies described in our manuscript provided a clear understanding of how this entire process works and what it entails. Therefore, participation in a clinical trial should be considered a separate procedure that does not impact the clinical process.

What do you think needs to change so that a diagnosis of HPV does not affect the marital relationship and the mental health of the patient?

Response: Although we did not come across any specific guidelines in our research, we believe that offering joint education for both the patient and her partner could be beneficial. This approach would ensure that both parties in the relationship receive sufficient information and are well-informed. We have added this issue to the manuscript (lines 531-534).

You have described the need for psychological support, but at what time and in what cases should it be involved in the management of the patient?

Response: We believe that psychological care should be provided immediately after receiving a Positive HR HPV Test and throughout the entire LEEP procedure, including preparations and post-surgery care. Training midwives in this area could be a prudent step. Moreover, providing informational and educational brochures to patients, allowing them to learn about the topic in the comfort of their homes, could also be a valuable solution. Furthermore, in addition to psychological care, sexological care should be provided upon request, as this may positively impact the sexual lives of women and their marital relationships. We have added a paragraph concerning to clarify this issue (lines 534-545).

In clinical practice, patients report being afraid of becoming pregnant after the history of HPv, but things are absolutely unrelated. Have you also investigated the relapse of these sexual dysfunctions also on the reproductive outcome? (That is, for example, that women who would have wanted a child avoided looking for it after being diagnosed with HPV).

Response: The topic of getting pregnant after HPV was not the purpose of this review, but it is certainly a very important problem, worthy of a separate paper. We have brought up the subject a bit in the manuscript (lines 448-452; 575-576).

Finally, for a multidisciplinary approach we propose to mention:

-DOI: 10.1186/s13027-022-00465-9

-DOI: 10.3390/jpm12091387

Response: We appreciate your suggestions and have added a new paragraph (lines 498-504; 525-528) to address above topics. Thank you for taking the time to review our manuscript.

Reviewer 2 Report

Dear Authors.

I found you manuscript The Impact of HPV diagnosis and the Electrosurgical Excision Procedure (LEEP) on mental health and sexual functioning:  A systematic review, very interesting and easy to follow.

The introduction section reveals good input of HPV generalities as well as patogenicity in general and in Poland.

I believe is enough information and is quality information.

In the methods section I saw that the authors used Prisma rules, they have the flow diagram, but also the observational studies were classified using the ROBINS-E tool.

I found this section well done and complete.

Now for the result section. In the beginning, when you fist see the 60 pages manuscript you may think it is too big, and it s true. But in the end when you start reading all the tables you realise that the information is presented very concise and it is actually useful to have it like that.

Table 2 presents  the Impact of HPV diagnosis on anxiety, depression and psychosocial quality of life, while Table three presents the impact of HPV diagnosis and LEEP procedure on depression/anxiety, quality of life and sexual function.

The conclusions presented in Table 4, about the Impact of the Loop Electrosurgical Excision Procedure (LEEP) on anxiety, depression, quality of life and sexual function, are very useful.

Now comes the Discussion section, which along with the conclusions gives good closure to the manuscript.

The limitation of the study are also presented.

So I may say the manuscript is well done, it may be published.

Congratulations!

Author Response

Thank you very much for your insightful evaluation and valuable feedback, we appreciate your time.

Dear Authors,

I found you manuscript The Impact of HPV diagnosis and the Electrosurgical Excision Procedure (LEEP) on mental health and sexual functioning:  A systematic review, very interesting and easy to follow. The introduction section reveals good input of HPV generalities as well as patogenicity in general and in Poland. I believe is enough information and is quality information. In the methods section I saw that the authors used Prisma rules, they have the flow diagram, but also the observational studies were classified using the ROBINS-E tool.I found this section well done and complete. Now for the result section. In the beginning, when you fist see the 60 pages manuscript you may think it is too big, and it s true. But in the end when you start reading all the tables you realise that the information is presented very concise and it is actually useful to have it like that.Table 2 presents  the Impact of HPV diagnosis on anxiety, depression and psychosocial quality of life, while Table three presents the impact of HPV diagnosis and LEEP procedure on depression/anxiety, quality of life and sexual function. The conclusions presented in Table 4, about the Impact of the Loop Electrosurgical Excision Procedure (LEEP) on anxiety, depression, quality of life and sexual function, are very useful. Now comes the Discussion section, which along with the conclusions gives good closure to the manuscript. The limitation of the study are also presented. So I may say the manuscript is well done, it may be published.

Congratulations!

Response: Thank you very much for your insightful evaluation and valuable feedback, we appreciate your time.

Reviewer 3 Report

The authors have embarked in an exhaustive literature review to address the potentially negative effects on women’s mental health and sexual functioning caused by the diagnosis and treatment of HPV related cervical lesions. The meticulous research and the inclusion of an abstract for each article unfortunately resulted in a manuscript which is too lengthy and perhaps more suitable for a Book Chapter.

Two drawbacks identified are: i) the non-inclusion of all publications (including those outside the principle databases) and that ii) it was not possible to conduct a formal meta-analysis, because of the heterogeneity of the articles.

The authors should consider outlining within a paragraph of the “Discussion” section an agenda of “Open-Pending” issues, to guide further research.

Some all too minor language polishing might be also required

Author Response

The authors have embarked in an exhaustive literature review to address the potentially negative effects on women’s mental health and sexual functioning caused by the diagnosis and treatment of HPV related cervical lesions. The meticulous research and the inclusion of an abstract for each article unfortunately resulted in a manuscript which is too lengthy and perhaps more suitable for a Book Chapter.

Two drawbacks identified are: i) the non-inclusion of all publications (including those outside the principle databases) and that ii) it was not possible to conduct a formal meta-analysis, because of the heterogeneity of the articles.

The authors should consider outlining within a paragraph of the “Discussion” section an agenda of “Open-Pending” issues, to guide further research.

Some all too minor language polishing might be also required

Response: Thank you for your important comments. As per your suggestion, we have addressed the open-pending issues regarding further research (lines 570-576). To shorten the article, all tables and figures have been moved to the "Supplemental Materials" section. Furthermore, the entire manuscript has undergone substantial English editing by a native speaker. We appreciate your time in reviewing our manuscript and providing valuable feedback.

Reviewer 4 Report

Sikorska et al. conducted a systematic review to investigate the impact of HPV diagnosis and subsequent treatment with the Electrosurgical Excision Procedure (LEEP) on patients' anxiety, depression, psychosocial quality of life, and sexual functioning. The review analyzed data from 60 records, including 50 papers on the impact of HPV diagnosis and 10 studies on the impact of LEEP procedure. The results indicate that HPV diagnosis has a negative impact on patients' psychosocial aspects, including the occurrence of depressive and anxiety symptoms, poorer quality of life, and sexual functioning in affected women. However, the impact of LEEP procedure on mental health and sexual life is not confirmed by the current studies, and further research in this area is necessary. The review highlights the need to implement additional procedures to minimize anxiety and distress in patients receiving a diagnosis of HPV or abnormal cytology and to improve awareness of sexually transmitted pathogens.

The claims are properly placed in the context of the previous literature. The experimental data support the claims. The manuscript is written clearly enough that most of it is understandable to non-specialists. The authors have provided adequate proof for their claims, without overselling them. The authors have treated the previous literature fairly. The paper offers enough details of methodology so that the experiments could be reproduced.

Comments

Nobody wants to read a manuscript which is 60 pages long. All figures and tables can be moved to "Supplemental".

Minor revisions

Page 2, line 88-91, "Treatment of such severe lesions involves excision of the pathological cone of the cervix known as Loop Electrosurgical Excision Procedure (LEEP) or Large Loop Excision of Transformation Zone (LLETZ)" => "Women with histologically confirmed high-grade lesions (CIN2+) in the cervical mucosa are typically recommended to undergo surgical treatment, which involves the removal of the affected area using a procedure known as Loop Electrosurgical Excision Procedure (LEEP) or Large Loop Excision of Transformation Zone (LLETZ)."

Add to the discussion:

Fear and anxiety caused by abnormal Pap smear results or positive HPV tests are largely due to the screening program itself (Jentschke 2020). Without screening, no women would know about asymptomatic precursors (CIN2+) to cervical cancer. Without screening, fewer women would be exposed to abnormal findings during screening, but more women would develop cervical cancer (Landy 2016).

The sensitivity of HPV testing surpasses that of cervical cytology (Pap test), but the specificity is lower. Primary HPV testing can reduce the incidence of cervical cancer more effectively than cervical cytology screening, albeit with a higher number of positive screening results. Given the negative emotional impact of screening, more specific tests are needed. The positivity rate of the HPV-test, the referral rate, and detection rates of CIN3+ can be influenced by the number of HPV types included in the test (Sørbye 2016). A previous study revealed that only six HPV types (16, 18, 31, 33, 45, or 52) accounted for 85% of invasive cervical cancer cases, whereas the other eight HPV types detected in a 14-type HPV DNA test were found in only 1.5% of invasive cervical cancer cases (Sundström 2021). It is worth noting that not all cases of CIN3 will progress to cancer, and only a minority of CIN2 cases will. Research indicates that if left untreated, only 30% of large CIN3+ lesions will develop into cervical cancer over 30 years (McCredie 2008), indicating that a considerable number of women with these conditions will undergo biopsy and, potentially, LEEP treatment needlessly. The primary objective of cervical cancer screening is not to detect as many CIN3+ cases as possible but to prevent as many cases of cervical cancer as possible while balancing benefits and harms (Malagón 2020).

References

Jentschke M, Lehmann R, Drews N, Hansel A, Schmitz M, Hillemanns P. Psychological distress in cervical cancer screening: results from a German online survey. Arch Gynecol Obstet. 2020 Sep;302(3):699-705. doi: 10.1007/s00404-020-05661-9. Epub 2020 Jun 27. PMID: 32594298; PMCID: PMC7447652.

https://pubmed.ncbi.nlm.nih.gov/32594298/

Sørbye SW, Fismen S, Gutteberg TJ, Mortensen ES, Skjeldestad FE. Primary cervical cancer screening with an HPV mRNA test: a prospective cohort study. BMJ Open. 2016 Aug 11;6(8):e011981. doi: 10.1136/bmjopen-2016-011981. PMID: 27515759; PMCID: PMC4985920.

https://pubmed.ncbi.nlm.nih.gov/27515759/

Sundström K, Dillner J. How Many Human Papillomavirus Types Do We Need to Screen For? J Infect Dis. 2021 May 20;223(9):1510-1511. doi: 10.1093/infdis/jiaa587. PMID: 32941611.

https://pubmed.ncbi.nlm.nih.gov/32941611/

McCredie MR, Sharples KJ, Paul C, Baranyai J, Medley G, Jones RW, Skegg DC. Natural history of cervical neoplasia and risk of invasive cancer in women with cervical intraepithelial neoplasia 3: a retrospective cohort study. Lancet Oncol. 2008 May;9(5):425-34. doi: 10.1016/S1470-2045(08)70103-7. Epub 2008 Apr 11. PMID: 18407790.

https://pubmed.ncbi.nlm.nih.gov/18407790/

Malagón T, Mayrand MH, Ogilvie G, Gotlieb WH, Blake J, Bouchard C, Franco EL, Kulasingam S. Modeling the Balance of Benefits and Harms of Cervical Cancer Screening with Cytology and Human Papillomavirus Testing. Cancer Epidemiol Biomarkers Prev. 2020 Jul;29(7):1436-1446. doi: 10.1158/1055-9965.EPI-20-0190. Epub 2020 Apr 24. PMID: 32332032.

https://pubmed.ncbi.nlm.nih.gov/32332032/

Landy, R., Pesola, F., Castañón, A. et al. Impact of cervical screening on cervical cancer mortality: estimation using stage-specific results from a nested case–control study. Br J Cancer 115, 1140–1146 (2016). https://doi.org/10.1038/bjc.2016.290

Author Response

Thank you very much for your insightful evaluation. As per your suggestion, all tables and most figures have been moved to the "Supplemental Materials" section, and the sentence mentioned in lines 88-91 has been edited accordingly. We have also added the indicated references to the discussion section (see lines 396-415). Thank you for taking the time to review our manuscript. We appreciate your valuable suggestions.

Round 2

Reviewer 1 Report

Thank you for revision.

For me the article is  now, ok.